# Effect of Drought Stress on Capsaicin and Antioxidant Contents in Pepper Genotypes at Reproductive Stage

**DOI:** 10.3390/plants10071286

**Published:** 2021-06-24

**Authors:** Tahir Mahmood, Rashid Mehmood Rana, Sunny Ahmar, Saima Saeed, Asma Gulzar, Muhammad Azam Khan, Fahad Masoud Wattoo, Xiukang Wang, Ferdinando Branca, Freddy Mora-Poblete, Gabrielle Sousa Mafra, Xionming Du

**Affiliations:** 1Department of Plant Breeding and Genetics, PMAS-Arid Agriculture University, Rawalpindi 46000, Pakistan; tahirmtaha@hotmail.com (T.M.); saimasaeedch@yahoo.com (S.S.); asmagul188@yahoo.com (A.G.); wattoo1619@gmail.com (F.M.W.); 2Institute of Biological Sciences, Campus Talca, Universidad de Talca, Talca 3465548, Chile; sunny.ahmar@yahoo.com; 3Department of Horticulture, PMAS-Arid Agriculture University, Rawalpindi 46000, Pakistan; drazam1980@uaar.edu.pk; 4College of Life Sciences, Yan’an University, Yan’an 716000, China; wangxiukang@yau.edu.cn; 5Department of Agriculture, Food and Environment (Di3A), University of Catania, 95123 Catania, Italy; branca@unict.it; 6Centro de Ciências Agrárias, Universidade Estadual da Região Tocantina do Maranhão, R. Godofredo Viana, 1300, Imperatriz 65901-480, MA, Brazil; gabrielle.smafra@yahoo.com.br; 7State Key Laboratory of Cotton Biology, Institute of Cotton Research, Chinese Academy of Agricultural Sciences (ICR, CAAS), Anyang 455000, China; duxiongming@caas.cn

**Keywords:** pepper, capsaicin, drought, reactive oxygen species, ROS, antioxidant, metabolomics

## Abstract

Pepper is one of the most important vegetables and spices in the world. Principal pungency is contributed by secondary metabolites called capsaicinoids, mainly synthesized in the placenta of pepper fruit. Various factors, including drought, limit pepper production. Flowering is one of the most sensitive stages affected by drought stress. The current study was conducted to determine the effect of drought on different pepper genotypes at the flowering and pod formation stages. Hot pepper (Pusajuala and Ghotki) and Bell pepper (Green Wonder and PPE-311) genotypes were subjected to drought (35% field capacity) at two different stages (flowering (DF) and pod formation (DP) stage). In comparison, control plants were maintained at 65% field capacity. The data regarding flowering survival rates, antioxidant protein activity, and proline content, were collected. Results indicated that parameters like flower survival percentage, number of fruits per plant, and fruit weight had significant differences among the genotypes in both treatments. A high proline level was observed in Green Wonder at the pod formation stage compared to other genotypes. Capsaicin contents of hot pepper genotypes were affected at the pod formation stage. Antioxidants like GPX were highly active (190 units) in Ghotki at pod formation. Bell pepper genotypes had a high APX activity, highly observed (100 units) in PPE-311 at pod formation, and significantly differ from hot pepper genotypes. In the catalase case, all the genotypes had the highest values in DP compared to control and DF, but Pusajuala (91 units) and Green Wonder (83 units) performed best compared to other genotypes. Overall, the results indicate that drought stress decreased reproductive growth parameters and pungency of pepper fruit as most of the plant energy was consumed in defense molecules (antioxidants). Therefore, water availability at the flowering and pod formation stage is critical to ensure good yield and pepper quality.

## 1. Introduction

Pepper (Capsicum spp.) is one of the important vegetables and spices of the world. It contains a wide range of phytochemicals, such as neutral and acidic phenolic compounds, which are important nutritional antioxidants that may reduce the risk of degenerative, mutagenic, and chronic diseases [1,2]. The phytochemicals in pepper have been reported to possess many biochemical and pharmacological properties, such as antioxidant, anti-inflammatory, anti-allergic, and anti-carcinogenic activities [3]. Ripened pepper is naturally rich in ascorbic acid (vitamin C) and provitamin A [4], which neutralizes free radicals in the human body and reduces the risk of diseases such as arthritis, cardiovascular disease, cancer and the aging process [5,6,7]. Carotenoids are fat-soluble antioxidants found in pepper and valuable for human epithelial cellular differentiation [8]. Furthermore, several studies have proved the antimicrobial activity of peppers [9,10].

Peppers are consumed either fresh, dried, or in powder form. Their high consumption may be ascribed to the unique pungency character contributed by secondary metabolites called capsaicinoids [11]. Capsaicin synthesis occurs via the phenylalanine ammonia-lyase (PAL) pathway by the action of several enzymes [12]. Capsaicin synthase (CS) is the main enzyme in capsaicinoids production. However, phenylalanine ammonia-lyase (PAL) plays an important role in stimulating capsaicin production compared to the rest of the enzymes. Condensing a molecule of vanillylamine (produced from phenylalanine) to a branched fatty acid (from 9 to 11 carbon atoms) generated from either valine or leucine yields capsaicinoids. Although there are more than 10 different capsaicinoid structures, capsaicin (CAP) and dihydrocapsaicin (DHCAP) are the most common, accounting for over 90% of all capsaicinoids [13]. The placenta of the fruit is the main region of capsaicin synthesis. However, capsaicin has also been extracted from pericarp and seed only because of their proximity to the placental area.

Changing climatic conditions at global and regional scales has threatened crop production by decreasing the availability of water resources [14,15,16,17]. Peppers are susceptible to water availability. Generally, water deficiency negatively influences dry matter accumulation, nutrient uptake, and crop yield [18,19]. Physiological, biochemical, and molecular changes occur in the plant during stress conditions. Low water availability affects metabolic processes such as the rate of photosynthesis, dry matter production, and yield. Oxidative damage of cellular organization occurs during drought stress through the production of reactive oxygen species. Plants may have a specific innate antioxidant mechanism to mitigate the effect of water stress. Specific physiological and molecular changes could make plants resistant to drought stress [20]. The increased proline content under drought stress usually indicates that plants adapt under stress conditions [21].

Moreover, water stress causes flower abscission (blossom drop) and smaller fruit size. Capsaicinoid accumulate at early fruit growth, reach maximum as fruits gain their final growth, and stop increasing as the fruit gains its maximum length [22]. Drought stress increased the pungency and ascorbic acid contents in mature fruits, although the pungency was also depends on genetic potential of different genotypes and stress duration. Drought boosted the activity of antioxidant enzymes in pepper leaves and fruits [23]. Jeeatid et al. recently disclose that appropriate drought stress could increase capsaicinoid contents in hot pepper [24]. Various enzyme activity increased in drought-stressed plants compared to control plants. PAL (Phenylalanine ammonia-lyase activity) is a crucial enzyme involved in capsaicinoid manufacture under drought stress, because its activity and capsaicinoid levels were significantly raised across the varied pungency levels of hot pepper cultivars [25]. Concentrations of several organic acids were higher under drought stress. Some cultivars had higher concentrations of ascorbic acid and total phenolic content under control and higher capsaicin and dihydrocapsaicin concentrations under drought stress [26].

The effect of drought stress on pungency level in pepper still is a debatable topic, ether it increases or decreases. There is evidence that capsaicin is upregulated and in some cased downregulated under drought stress conditions; this depends on the genetic makeup of cultivars and the stress levels. Thus, the effect of drought stress during capsaicin accumulation could be critical and hence needs investigation. Moreover, antioxidant machinery and proline accumulation during stress may play a critical role in stress tolerance. Therefore, the current study was aimed at addressing these points where drought stress was applied at two stages of reproductive development, plants were then evaluated for capsaicin content, antioxidant activity, proline content, flower drop, fruit weight, and fruit shape index.

## 2. Materials and Methods

### 2.1. Plant Material

The field experiment was conducted at the Department of Plant Breeding and Genetics, Pir Mehr Ali Shah Arid Agriculture University Rawalpindi, Pakistan. Pepper genotypes viz. Pusa Juala, Ghotki, Green Wonder, and Hybrid-311 were used in the experiment. The nursery was established and transplanted in plastic pots (6″ × 8″) containing peat: sand: topsoil (1:1:1). The experiment was designed in a completely randomized design (CRD) with three replications. The plants were subjected to drought at two growth stages, i.e., flowering stage (DF; early floral bud stage) and pod formation stage (DP), by maintaining field capacity at 35%. The control plants were watered regularly to maintain a field capacity of 65%. Plants of stress treatments were kept under drought till maturity/harvesting.

### 2.2. Flower and Fruit Data

The data regarding the production traits such as number of flowers was observed by counting the total flowers produced before the application of drought. The number of flowers dropped after drought stress application was also counted to calculate flower survival percentage (FS). Mature fruits were harvested from pepper plants of both control and treatment conditions. Fruits were harvested at 45 DAF (days after flowering). Fruit growth was estimated by recording data like counting the total number of fruits produced per plant (FPP), fruit fresh weight (FFW), fruit dry weight (DFW) (oven-dried) was measured by weighing balance. The length of the fruit was measured as polar diameter, and width was measured in equatorial diameter. The shape index was then calculated by dividing polar diameter by the low diameter using the following formula.

Shape index (FSI) = polar diameter/equatorial diameter.

For longer fruits, the range of shape index is greater than one, but it equals one [27] for round fruits.

### 2.3. Detection of Capsaicin Contents by Scoville Scale

Capsaicin contents were estimated by performing an organoleptic test. The weighed (2.5 g) pepper samples and then ground the samples by adding 2.5 mL of 95% ethanol in a pestle and mortar. The dilutions were then made as 1:10, 1:100, 1:1000, 1:10,000, and 1:100,000 using 5% sucrose. In accordance with the Scoville Heat Unit, organoleptic testing was performed with a panel of 4 people to assess pungency levels [27].

### 2.4. Proline Determination

Proline content of pepper was determined using the method proposed by Bates et al., [28]. At first, ninhydrin reagent was prepared in such a way so that it was utilized for proline estimation within two hours of preparation. The samples were ground with a pestle and mortar and homogenized in 2 mL of 3% sulphosalicylic acid. The samples were then centrifuged at 8042.4× *g* for 10 min. The upper phase was aspirated in a separate test tube. The extract (1 mL; containing 25 mg ninhydrin) was mixed with 1 mL of a mixture of glacial acetic acid and 6 M orthophosphoric acid (3:2 *v*/*v*). The mixture was then boiled at 100 °C for 60 min in a water bath in the tube covered with aluminum foil to prevent excess evaporation. The reaction was terminated by putting the tubes in an ice bath quickly, and 5 mL toluene was added using a dispenser. Each tube was then shaken vigorously for 15 to 20 s in an electrical shaker and the layer was allowed to separate for 30 min. The upper phase of the reaction mixture was used to determine proline by observing absorbance at 520 nm in a spectrophotometer (Spekol 1300). A sample containing toluene (without reaction mixture) was used as blank. The proline content was estimated by the standard curve method (Appendix A), where proline concentration is mentioned as µmoles proline/g of fresh weight [28].

### 2.5. Antioxidant Enzyme Assay

Enzymes were extracted from pepper samples by homogenizing 100 mg fresh pepper in 1 mL of phosphate buffer. Homogenates were centrifuged at 893.6× *g* at 4 °C for 10 min to remove plant debris [29]. Catalase (CAT) activity was determined by measuring the absorbance of reaction (334 µL enzyme extract + 666 µL 73 mM H_2_O_2_) at 240 nm for 3 min [30] and measured as mM H_2_O_2_ μg/min × g FW (FW: fresh weight). Ascorbate peroxidase (APX) activity was determined by measuring the absorbance of reaction (450 µL enzyme extract + 100 µL 17 mM H_2_O_2_: 25 mM ascorbate) at 290 nm for 3 min and measured as μg ascorbic acid μg/min × g of FW. Guaiacol peroxidase (GPX) activity was determined by measuring the absorbance of reaction (450 µL enzyme extract + 100 µL 17 mM H_2_O_2_: 2% guaiacol) at 510 nm 3 min. GPX was measured as μmol of tetraguaiacol μg/min × g of FW. One unit of enzyme activity was defined as a decrease (CAT and APX) or increase (GPX) in the relative Activity.

### 2.6. Statistical Analysis

Analysis of variance (ANOVA) as described by Steel et al. [31] was done for all the attributes, and the least significant difference (LSD) was used to assess the difference among genotypes at different treatments. JMP^®^ (Version (15.0), SAS Institute Inc., Cary, NC, USA, 1989–2019) was used to do hierarchical clustering, dendrogram correlation, and constellation plot.

## 3. Results

### 3.1. Clustering and Comparative Analysis of Pepper Genotypes and Traits

Hierarchical clusters and dendrogram heat map of pepper genotypes and traits, including antioxidants and production traits, are shown in Figure 1. The pepper genotypes are grouped into three major clusters under three different treatments, including control, D(F), and D(P), which showed the significant effects of treatment application. Additionally, the measured traits, including antioxidants and production traits, are divided into two major groups. This indicates the unique response of hot pepper and bell pepper to the drought stress at different reproductive stages.

### 3.2. Effect of Drought Stress on Antioxidants and Production Traits

Results indicated the different responses of hot pepper and bell pepper to drought stress at different reproductive stages for different antioxidant and production traits. Both genotypes of hot pepper had higher FSI, GPX, and catalases at fruiting stages under drought stress. In contrast, the bell had a higher magnitude of catalases APX and proline contents under drought stress conditions, which is more significant in Green Wonder at D(P) stage (Figure 1).

### 3.3. Association of Production Traits with Antioxidants

The association among the different traits indicates a common response to those traits that may have involved a similar responsive mechanism under a specific condition. In the present study, we observed a strong positive correlation between FS and FFW, followed by DFW and FFW. Meanwhile, in the antioxidants, a positive and strong association was noticed between catalase and APX, followed by APX with GPX, and catalase with GPX, respectively. A negative correlation was observed in FSI and GPX with DFW. The results indicated that most of the production traits had a negative or weak correlation with antioxidants, indicating that the pepper plant compromises the production of antioxidants under drought stress to overcome the drought stress (Figure 1).

### 3.4. Principal Component Analysis

A multivariate, principal component analysis (PCA) was performed to explain how drought stress affected analyzed parameters and the relationship among them (Figure 2). PCA explained overall about 67% of the variation and differences among genotypes, treatments, and traits. 3D biplot explained the effect of drought stress on production traits and antioxidants at D(F) and D(P) stages. Additionally, biplot showed the significant differences among three clusters and groups of pepper genotypes under three different conditions, which also explained the three clusters and their responses to drought stress in dendrogram and hierarchical clustering (Figure 1).

### 3.5. Mean Performance and Graphical Presentation of Traits

#### 3.5.1. Flower Survival Percentage

Flower production in pepper plants was recorded carefully under each treatment, such as control, DF, and DP. Plants receiving limited water supply before the flowering suffered delayed and reduced flowering. The maximum number of flowers produced reached 71–199 flowers per plant, varying among genotypes. However, the production of the flowers in control conditions differed significantly from the flower production rates of the stressed plants. ANOVA results depicted significant variation for flowering data, indicating that the flower and pod formation variation was due to the treatment effects. Hot pepper genotype Ghotki had a higher survival percentage (55%) than other genotypes under control conditions. Pusajuala and Ghotki showed a significant reduction in both treatments as compared to control. Bell pepper genotypes like Green Wonder and PPE-311 showed less reduction than control but significantly differed with the hot pepper genotypes (Figure 3).

#### 3.5.2. Number of the Fruits Per Plant

The extent to which fruit number per plant decreases depends on the development stage at which the plant faces stress. Water deficiency at the vegetative stage affects fruit yield and results in stunted plants. Results showed that fruit number was significantly higher in control plants than DF or DP (Appendix A). Hot pepper genotype Pusajuala gave higher fruit production (70–92 fruits per plant) than other varieties like Ghotki, Green wonder, and PPE-311 under control conditions. Ghotki and Green wonder showed maximum reduction in DF and DP compared with control. Still, Pusajuala performed much better under DF and DP in terms of fruit number, whereas bell pepper variety “Green wonder” produced fewer fruits than PPE-311 (Figure 4). In terms of control, hot pepper had a maximum number of fruits per plant compare to bell pepper.

#### 3.5.3. Fruit Weight

Stress application at flowering and pod formation stage follows a decreasing trend for all pepper genotypes’ fresh and dry fruit weights except PPE-311 showing a slight increase in dry weight under DF. Generally, results indicate that fruit from DF weighed more as compared to the fruits in DP. However, plants performed best only at optimum water level (control). ANOVA of DF (Appendix A) was significant, showing that those genotypes can perform better in terms of fruit weight if they receive water stress only at flowering. Bell pepper genotypes performed better and showed significant results as compared to hot pepper genotypes. Green wonder peppers had the highest fruit weight (10.63 g) under control conditions (Figure 5).

#### 3.5.4. Proline Contents

Levels of proline varied significantly among genotypes under different treatments. At the pod formation stage (DP), the highest proline levels were observed in Pusajuala, Ghotki, Green Wonder, and PPE-311. Significant results indicated that proline amounts continue to increase in all varieties under drought stress treatments compared to control. Bell pepper genotypes showed a significant increase in both treatments as compared to Hot pepper genotypes. Maximum proline level was observed in Green Wonder at pod formation stage as compared to other genotypes. Ghotki ranks second, and the minimum level was observed in Pusajuala and PPE-311 at D(P) (Figure 6).

#### 3.5.5. Capsaicin Contents

A change in pungency between stressed and control plants was observed only for the hot pepper’s genotypes (Pusajuala and Ghotki). In contrast, bell peppers (Green Wonder and PPE-311) did not show any change in their so-called slightest pungency. In control conditions, Pusajuala gave us approximately 10,000 SHU, whereas Ghotki gave 10,000 SHU. Green wonder and PPE-311 gave only ten Scoville units in all three treatments. In DP, fruits show a drastic decrease in pungency units of hot pepper genotypes (Appendix A). Stress application at this stage (pod formation) may restrict capsaicin synthase activity that might have affected the pungency of these hot pepper genotypes.

#### 3.5.6. Antioxidant Enzyme Activity

Catalase and peroxidase activity is associated with the antioxidative properties of the peppers. GPX, APX, and CAT are some important enzymes related to the scavenging of free radicals. Antioxidant activity is more dependent on fruitage. The activity of antioxidants was found significantly less in DF than in control plants (Figure 7). However, a significant increase in peroxidase activity was observed in DP compared to the control and DF fruits. Maximum activity of GPX (190 units) was observed in Ghotki as compared to other genotypes. Bell pepper genotypes had a high APX activity, highly regarded (100 units) in PPE-311 at pod formation, and significantly differed from hot pepper genotypes. In the catalase case, all the genotypes had the highest values in DP compared to control and DF, but as a whole Pusajuala (91 units) and Green Wonder (83 units) performed best compared to other genotypes.

## 4. Discussion

Pepper plants are known to produce a very immediate response when they are subjected to any stress. If moisture contents in soil decrease, pepper plants begin to wilt immediately but never wilt permanently and regain their normal state upon water supply [32]. Abiotic stresses urge the production of reactive oxygen species (ROS), which cause the oxidation of carbohydrates, lipids, proteins, and DNA [33,34]. Various antioxidants maintain the balance between ROS production and their scavenging events. Similarly, green pepper has higher antioxidative enzymes that protect against harmful ROS and free radicals in our results. The quality of green pepper depends on the antioxidants’ content, as reported recently [35,36].

The amino acid proline, a quaternary amine, is probably the most common compatible solute synthesized by plants to respond to abiotic stress [16,37,38,39]. It plays a critical role in osmoregulation as well as acting as low-molecular-weight chaperons. Proline has also been reported to play an important role in detoxification of ROS and the regulation of gene expression as a signaling molecule [40,41]. In the present study, the levels of proline accumulation varied among varieties at the flowering and pod formation stages. The proline content was increased at the pod formation stage in all the genotypes. Kaur et al. [42] reported that chickpea genotypes that performed better under drought showed significant proline levels than those of genotypes that were sensitive under water deficit conditions. Under the stressed condition, proline is synthesized from glutamate and adjusts the osmoregulation at the cellular level. It may also improve the activity of antioxidant enzymes, which act as the protective mechanism against stress conditions. Sikha et al. [43] also reported increased proline content in leaves and roots than control in *Capsicum annuum* Solan Bharpur during PEG and NaCl-induced stress. Our findings suggest that, at the pod formation stage, proline production is significantly high, which would be helpful to mitigate the effect of drought stress at pod formation and may stabilize fruit weight.

Capsaicin and other related compounds, commonly called capsaicinoids, are phenolic compounds characteristic of some fruits of the genus Capsicum [44]. Capsaicin and dihydrocapsaicin are the most abundant capsaicinoids in hot peppers [45] and are responsible for 90% of their pungency [46,47]. In the current study, Pusajuala was the most pungent pepper variety with thin and longer fruits. The pungency (Scoville Heat Unit) of pepper genotypes was not affected by drought at the flowering stage as the capsaicin contents have remained unchanged in all the genotypes. These results suggest that the flowering stage might not be sensitive to the accumulation of the capsaicin contents. This observation follows many studies claiming that capsaicin accumulation starts at about 10 DAF [12]. Capsaicin biosynthesis is faster, and its accumulation starts upon the formation of the pod. Our results showed that the plants exposed to drought at the pod formation stage had a considerable decrease in capsaicin contents. However, previous studies reported a significant increase in capsaicin contents under drought [12]. There are contradictory views on the pungency of pepper genotypes in water stress conditions. Some studies claim that pungency increases under water stress [48], but others gave opposite views [32].

Similarly, capsaicin concentration varies among different genotypes. This contrast could be due to a short spell of stress at the pod formation stage in the current study, while other prolonged drought stress was used in reports. The stress at the onset of capsaicin biosynthesis might have interrupted initial levels of capsaicin, which could not be recovered afterward. However, a detailed investigation is required to justify these speculations.

ROS accumulation has been proposed to damage the morphological structure and physiological metabolism of plants. To keep excess ROS under homeostatic control, plants have antioxidative enzymes and metabolites, including catalase (CAT), guaiacol peroxidase (GPX), and ascorbate peroxidase (APX) [49]. In the current study, peroxidases (GPX, APX) were decreased at flowering stage compared to the control. However, an increase in the activity of these enzymes was observed during stress at the pod formation stage. A decrease in capsaicin contents in pods that experienced drought during the pod formation stage could be explained by increased peroxidases, as Iwai et al. [45] suggested the involvement of peroxidases in capsaicinoid degradation. Our findings indicate that if a plant faces drought stress at the pod formation stage, the antioxidative enzymes scavenge the ROS, including capsaicin content, making this stage less sensitive than flowering regarding yield; however, its capsaicin contents would be affected.

Morphological traits like fruit production, flower survival percentage, and fruit shape index decreased significantly at the flowering and pod formation stage due to water stress conditions. Jaimez et al. [50] also observed a reduction in the number of flowers and a delay in the occurrence of maximum flowering in response to water stress on fruit production in *C. chinense*. Fruit diameter was also reduced in pears due to water stress compared to control plants, because of osmotic adjustment in fruit [51]. Similar studies also observed a reduction in fruit weight in pepper at pod formation compared to water stress applied at the flowering stage.

## 5. Conclusions

Peppers are one of the most consumed vegetables in the world. Water stress decreases reproductive growth parameters and pungency of pepper fruit, whereas antioxidant activity was significantly increased in the fruits harvested 45 days after flowering (DAF). The performance of all these traits is dependent on the fruitage as well as on the environmental conditions. The different response of hot pepper and bell pepper to drought stress at different reproductive stages was observed, which may result from relatively different drought coping mechanisms opted for by different pepper species. The production traits had a negative and weak correlation with antioxidants, indicating that the pepper plant compromises antioxidant production under drought stress to overcome the drought stress.

## Figures and Tables

**Figure 1 plants-10-01286-f001:**
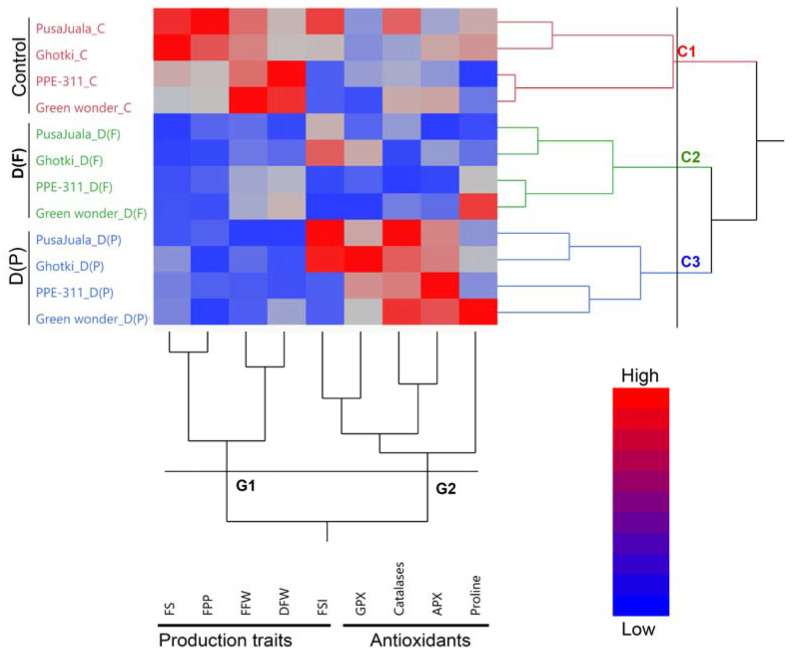
Hierarchical clusters and dendrogram of pepper genotypes and studied traits. Hierarchical clusters indicating three major clusters, including C1, C2, and C3 of pepper genotypes, also indicate two major groups (G1 and G2) of studied triat.

**Figure 2 plants-10-01286-f002:**
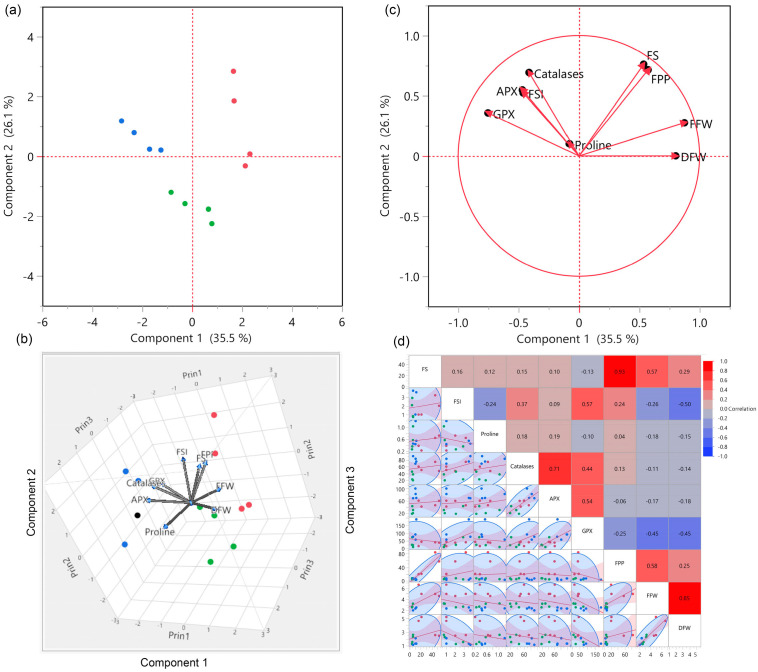
PCA, biplot, and correlogram of antioxidants and production traits in four hot and bell pepper genotypes under three different environments. (**a**–**c**) biplot distribution of four pepper genotypes under three different conditions. (**d**) correlogram of antioxidants and production traits. According to the color scheme, red indicates the cluster of genotypes under control, green under D(F), and blue under D(P).

**Figure 3 plants-10-01286-f003:**
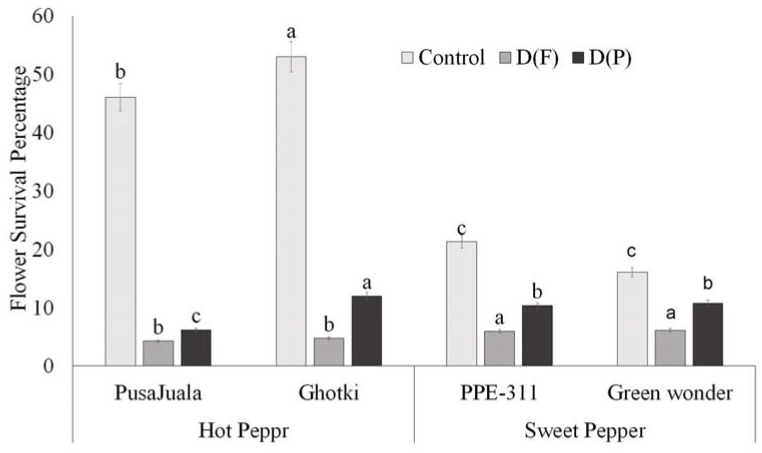
Flower survival percentage. Means of three repetitions for each genotype with the same letters indicate the same means, and different letters indicate significant differences. Vertical bars represent the standard error of means (bars = ±se). Bars without letters represent non-significant means (LSD *p* < 0.05). DF = water stress at the flowering stage, DP = water stress at the pod formation stage.

**Figure 4 plants-10-01286-f004:**
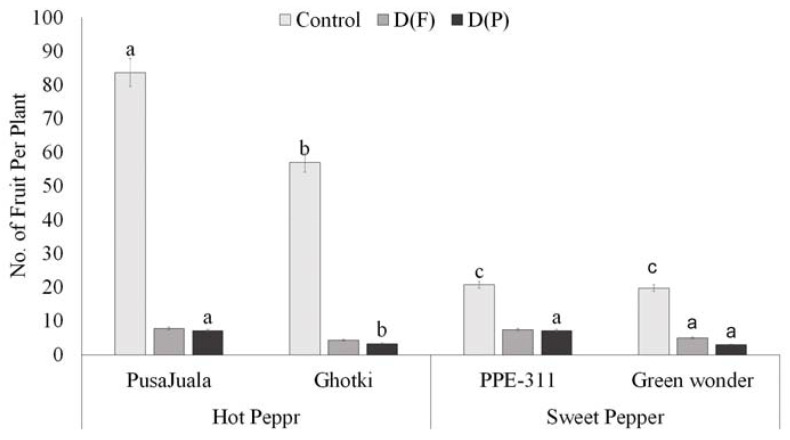
The number of fruits per plant. Means of three repetitions for each genotype with the same letters indicate the same means, and different letters indicate significant differences. Vertical bars represent standard error of means (bars = ±se) (LSD *p* < 0.05). DF = water stress at the flowering stage, DP = water stress at the pod formation stage.

**Figure 5 plants-10-01286-f005:**
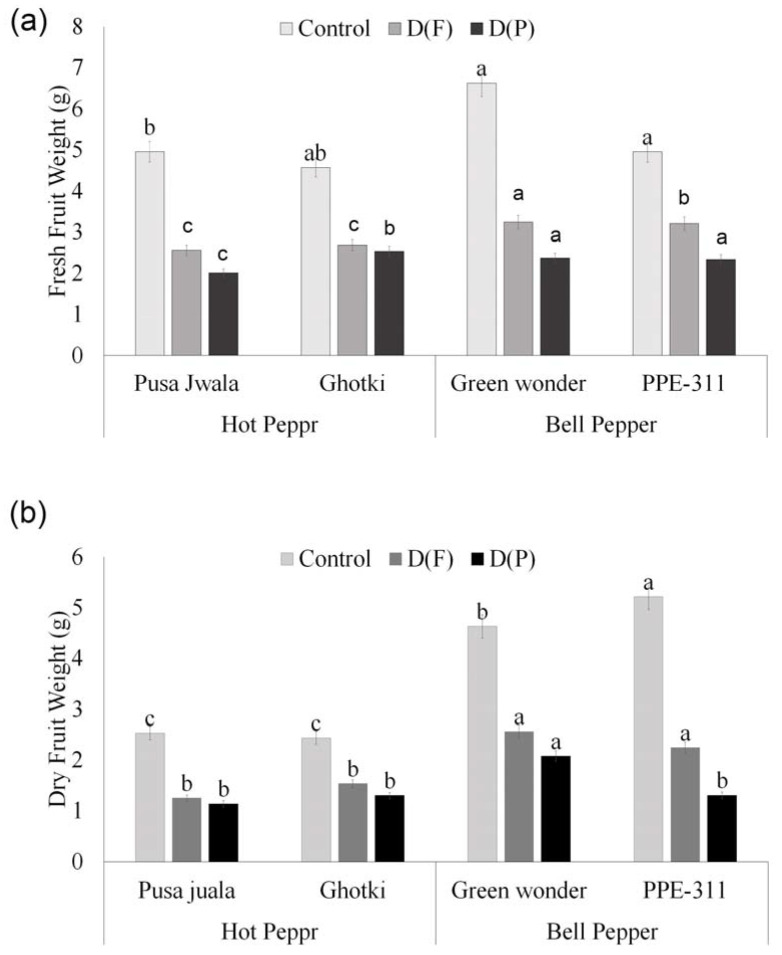
Fresh (**a**) and dry (**b**) fruit weight. Means of three repetitions for each genotype with the same letters indicate the same means, and different letters indicate significant differences. Vertical bars represent the standard error of means (bars = ±se). DF = water stress at flowering stage, DP = water stress at the pod formation stage.

**Figure 6 plants-10-01286-f006:**
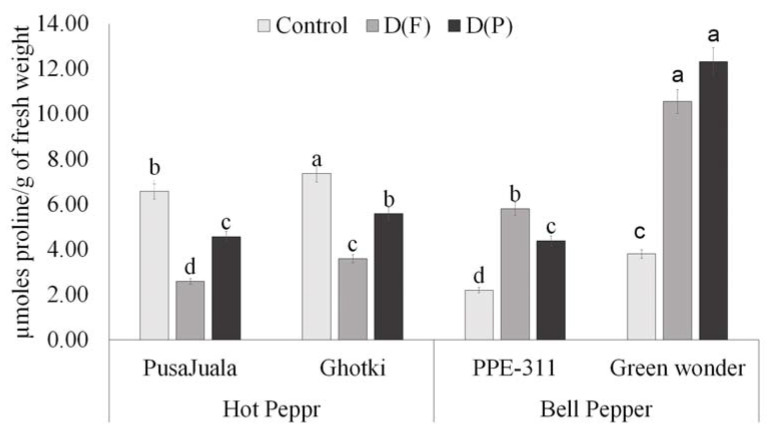
Proline contents of chili fruits. Means of three repetitions for each genotype with the same letters indicate the same means, and different letters indicate significant differences. Vertical bars represent standard error of means (bars = ±se) (LSD *p* < 0.05). DF = water stress at flowering stage, DP = water stress at the pod formation stage.

**Figure 7 plants-10-01286-f007:**
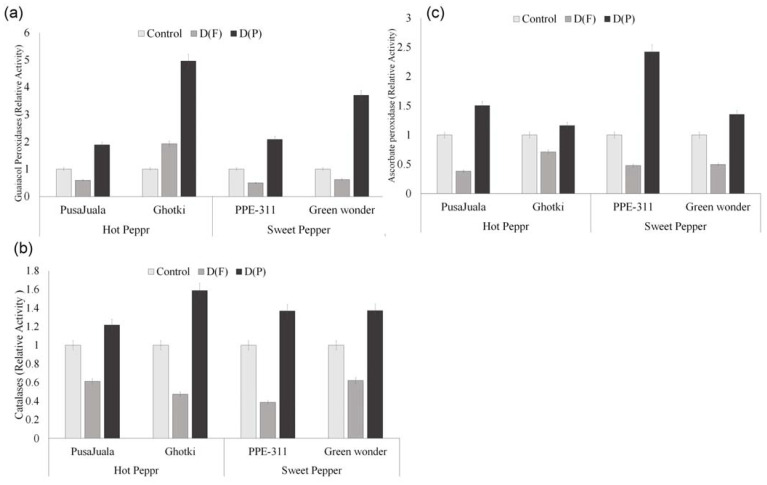
Ascorbate peroxidases (APX), catalase (CAT), and guaiacol peroxidases (GPX) activity of chili fruits. Means of three repetitions for each genotype with the same letters indicate the same means, and different letters indicate significant differences. Vertical bars represent standard error of means (bars = ±se) (LSD *p* < 0.05). DF = water stress at flowering stage, DP = water stress at the pod formation stage.

## Data Availability

Not applicable.

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
