# Peer review of "Effect of Drought Stress on Capsaicin and Antioxidant Contents in Pepper Genotypes at Reproductive Stage"

_plants, 2021, doi:10.3390/plants10071286_

Round 1

Reviewer 1 Report

Major comments

There are some major comments that need to be addressed before final assessment of a manuscript. The paper is lacking mention of research done by others who investigated drought stress in pepper and how it relates to capsaicin content and physiological responses in plants. These should be noted in the Introduction and/or discussed in the Discussion section of the manuscript.

Some examples:

  1. Effect of Water Supply on Physiological Response and Phytonutrient Composition of Chili Peppers

By: Agyemang Duah, Stella; Souza, Clarice Silva e; Nagy, Zsuzsa; et al.

WATER   Volume: ‏ 13   Issue: ‏ 9     Article Number: 1284   Published: ‏ MAY 2021

  1. Influence of irrigation on yield and primary and secondary metabolites in two chilies species, Capsicum annuum L. and Capsicum chinense Jacq

By: Zamljen, Tilen; Zupanc, Vesna; Slatnar, Ana

AGRICULTURAL WATER MANAGEMENT   Volume: ‏ 234     Article Number: 106104   Published: ‏ MAY 1 2020

  1. Factors affecting the capsaicinoid profile of hot peppers and biological activity of their non-pungent analogs (Capsinoids) present in sweet peppers

By: Uarrota, Virgilio Gavicho; Maraschin, Marcelo; de Bairros, Angela de Fatima M.; et al.

CRITICAL REVIEWS IN FOOD SCIENCE AND NUTRITION   Volume: ‏ 61   Issue: ‏ 4   Pages: ‏ 649-665   Published: ‏ FEB 21 2021

  1. Screening of Chilli Pepper Genotypes as a Source of Capsaicinoids and Antioxidants under Conditions of Simulated Drought Stress

By: Kopta, Tomas; Sekara, Agnieszka; Pokluda, Robert; et al.

PLANTS-BASEL   Volume: ‏ 9   Issue: ‏ 3     Article Number: 364   Published: ‏ MAR 2020

  1. Influence of water stresses on capsaicinoid production in hot pepper (Capsicum chinense Jacq.) cultivars with different pungency levels

By: Jeeatid, N.; Techawongstien, S.; Suriharn, B.; et al.

FOOD CHEMISTRY   Volume: ‏ 245   Pages: ‏ 792-797   Published: ‏ APR 15 2018

  1. Physiological response of the three most cultivated pepper species (Capsicum spp.) in Africa to drought stress imposed at three stages of growth and development

By: Okunlola, Gideon Olarewaju; Olatunji, Olusanya Abiodun; Akinwale, Richard Olutayo; et al.

SCIENTIA HORTICULTURAE   Volume: ‏ 224   Pages: ‏ 198-205   Published: ‏ OCT 20 2017

  1. Enzymatic Changes in Phenylalanine Ammonia-Iyase, Cinnamic-4-hydroxylase, Capsaicin Synthase, and Peroxidase Activities in Capsicum under Drought Stress

By: Phimchan, Paongpetch; Chanthai, Saksit; Bosland, Paul W.; et al.

JOURNAL OF AGRICULTURAL AND FOOD CHEMISTRY   Volume: ‏ 62   Issue: ‏ 29   Special Issue: ‏ SI   Pages: ‏ 7057-7062   Published: ‏ JUL 23 2014

  1. Impact of Drought Stress on the Accumulation of Capsaicinoids in Capsicum Cultivars with Different Initial Capsaicinoid Levels

By: Phimchan, Paongpetch; Techawongstien, Suchila; Chanthai, Saksit; et al.

HORTSCIENCE   Volume: ‏ 47   Issue: ‏ 9   Pages: ‏ 1204-1209   Published: ‏ SEP 2012

The paper has several methodological issues. The amount of capsaicin is not determined quantitatively (HPLC or similar) but estimated by organoleptic test, which is a limitation of the study that should be mentioned in the discussion. Proline content was determined by ninhydrin reagent, but it seems without the proline standard was not used to assess the content quantitatively? This should also be noted. How are enzyme units calculated for each antioxidant enzyme and expressed? (Per g of fresh weight?)

Figure 1. needs a clear explanation. This reviewer has done some guessing to elucidate which production traits are behind the abbreviations in the Fig. 1. They do not seem to be explained anywhere in the rest of the text.  

Lines 194-197: The explanation in the text seems to contradict the respective Figure 3. (Ghotki better in terms of fruit number than Green wonder or PPE-311?)

Lines 221-222: It is not clear how the sentence relates to the Figure 5, i.e. it contradicts the explanation in 225-227?

Capsaicin contents: There is some data missing??? The text refers to ‘drastic decrease in pungency units’ in DP, but there are no data to document this.

Minor comments

Line 24: 'etc' should not be used here. Please, be specific about the parameters measured.

Line 27: '1.1. absorbances at 520 nm' is not a quantitative measure of the proline level and should be omitted from the abstract

Line 44: Antioxidant activity was described in the previous sentence, needs not be repeated here.

Line 47: It is a kind of overstatement to claim that vitamin C and provitamin A reduce the risk of cancer (and some other diseases); ref. 8? please check if ref. 8 is appropriately cited to support the role in epithelial differentiation or the role in cancer prevention.

Line 53: Capsaicin biosynthesis pathway should be described more clearly – there are no two pathways for capsaicin synthesis, instead products of two metabolic pathways are needed for the action of capsaicin synthetase…A reference is missing for the important role of PAL.

Line 104: Please provide full reference for Peterson, 1959. 'For longer fruits, the shape index is greater then one, but is equal to one'. The sentence is not clear.

Line 111: Please provide full reference for Reddy and Sasikala, 2013

Line 134: Please provide full reference for UA EPA, 1994.

Line 189: ‘Water deficiency at the vegetative stage does not affect fruit yield.’ If this is a general knowledge and not the observation form the study, it should be moved from the Results to the Discussion section.

Figure 4. ‘Sweet pepper’ used in the figure instead of bell pepper

List of references needs to be edited for formatting and consistency.

Author Response

Dear Reviewer,

Thank you for help us to improve our MS.

Major comments

There are some major comments that need to be addressed before final assessment of a manuscript. The paper is lacking mention of research done by others who investigated drought stress in pepper and how it relates to capsaicin content and physiological responses in plants. These should be noted in the Introduction and/or discussed in the Discussion section of the manuscript.

Point 1. The paper is lacking mention of research done by others who investigated drought stress in pepper and how it relates to capsaicin content and physiological responses in plants. These should be noted in the Introduction and/or discussed in the Discussion section of the manuscript.

Response: Thanks for valuable suggestions. We have explained that how capsaicin content and physiological responses under drought stress in plants. Additionally, we have discussed the cross talk on effects of drought stress on capsaicin content in pepper, in the introduction section.

Point 2. The paper has several methodological issues. The amount of capsaicin is not determined quantitatively (HPLC or similar) but estimated by organoleptic test, which is a limitation of the study that should be mentioned in the discussion. Proline content was determined by ninhydrin reagent, but it seems without the proline standard was not used to assess the content quantitatively? This should also be noted. How are enzyme units calculated for each antioxidant enzyme and expressed? (Per g of fresh weight?)

Response: Yes, the amount of capsaicin is estimated by organoleptic test because the HPLC facility was not available that time, but we used a standard method to measure the pungency level which has been using in the previous studies.

We re-considered the proline estimation method from 520nm absorbance values by standard curve method.  The figure 4 has revised in MS.  where proline concentration is mentioned as µmoles proline/g of fresh weight.

The unit of enzyme activity has defined as a decrease (CAT and APX) or increase (GPX) in the absorbance of 0.001/min at respective wavelength.

Point 3. Figure 1. needs a clear explanation. This reviewer has done some guessing to elucidate which production traits are behind the abbreviations in the Fig. 1. They do not seem to be explained anywhere in the rest of the text.  

Response: Production traits have been explained in section 2.2

Point 4. Lines 194-197: The explanation in the text seems to contradict the respective Figure 3. (Ghotki better in terms of fruit number than Green wonder or PPE-311?)

Response: The related explanation has been revised to make it clearer and more precise in last three line of section 3.2.2

Point 5. Lines 221-222: It is not clear how the sentence relates to the Figure 5, i.e. it contradicts the explanation in 225-227?

Response: Figure 5 has been replaced and explanation also has revised in section 3.2.4

Point 6. Capsaicin contents: There is some data missing??? The text refers to ‘drastic decrease in pungency units’ in DP, but there are no data to document this.

Response: Thanx for your concern supplementary data has been added to support our result and discussion.

Minor comments

Line 24: 'etc' should not be used here. Please, be specific about the parameters measured.

Response: Addressed (line28)

Line 27: '1.1. absorbances at 520 nm' is not a quantitative measure of the proline level and should be omitted from the abstract

Response: Addressed (line 31)

Line 44: Antioxidant activity was described in the previous sentence, needs not be repeated here.

Response:  In the second sentence we explained the types of phytochemicals so anti-oxidants is one of them

Line 47: It is a kind of overstatement to claim that vitamin C and provitamin A reduce the risk of cancer (and some other diseases); ref. 8? please check if ref. 8 is appropriately cited to support the role in epithelial differentiation or the role in cancer prevention.

Response:  Yes, author appropriately explained it in a review article titled: “vitamin C, and vitamin E as protective antioxidants in human cancers”, published in Annual Review of Nutrition

Line 53: Capsaicin biosynthesis pathway should be described more clearly – there are no two pathways for capsaicin synthesis, instead products of two metabolic pathways are needed for the action of capsaicin synthetase…A reference is missing for the important role of PAL.

Response: Capsaicin biosynthesis pathway has explained more clearly.

Line 104: Please provide full reference for Peterson, 1959. 'For longer fruits, the shape index is greater then one, but is equal to one'. The sentence is not clear.

Response:  Reference added and sentence also has revised

Line 111: Please provide full reference for Reddy and Sasikala, 2013

Response:  Added

Line 189: ‘Water deficiency at the vegetative stage does not affect fruit yield.’ If this is a general knowledge and not the observation form the study, it should be moved from the Results to the Discussion section.

Response: it was misleading statement, it has revised

Figure 4. ‘Sweet pepper’ used in the figure instead of bell pepper

Response: Figure revised

List of references needs to be edited for formatting and consistency.

Response: Addressed

Reviewer 2 Report

The manuscript of Mahmood and colleagues is particularly interesting and original. The authors discuss the link between flowering environmental conditions and capsaicin content in Capsicum species. The authors highlight how drought in particular can affect the content of secondary metabolites of the plant. This study is very important for potential agronomic applications in the exploitation of capsaicin production.

The manuscript is clear, and a correct statistical approach has been applied. I suggest to the authors two references that could enrich their manuscript:

Mahdavi A et al. Variation in Terpene Profiles of Thymus vulgaris in Water Deficit Stress Response. Molecules. 2020

Naservafaei S et al. Biological Response of Lallemantia iberica to Brassinolide Treatment under Different Watering Conditions. Plants (Basel). 2021

Author Response

Thanks for valuable comment.

The manuscript of Mahmood and colleagues is particularly interesting and original. The authors discuss the link between flowering environmental conditions and capsaicin content in Capsicum species. The authors highlight how drought in particular can affect the content of secondary metabolites of the plant. This study is very important for potential agronomic applications in the exploitation of capsaicin production.

The manuscript is clear, and a correct statistical approach has been applied. I suggest to the authors two references that could enrich their manuscript:

Mahdavi A et al. Variation in Terpene Profiles of Thymus vulgaris in Water Deficit Stress Response. Molecules. 2020

Naservafaei S et al. Biological Response of Lallemantia iberica to Brassinolide Treatment under Different Watering Conditions. Plants (Basel). 2021

Response: Thank you for your response. Suggesting articles has been citied

Reviewer 3 Report

My comments are included in the PDF file

Author Response

Dear Reviewer,

Thanks for the help to improve the MS.

Point 1: I would be cautious in making such a conclusion. Yes, paprika is popular, but it is not the most important crop.

Response: We agreed and revised the statement (Line 40)

Point 2: No analyzes were made to confirm the stress of drought. It is understood that the authors assume that the lack of watering caused drought stress, but this should be confirmed nevertheless. There are many methods for assessing the impact of stress factors (e.g plant photosynthesis or plant chlorophyll fluorescence)

Response: There is no direct method is available to measure drought tolerance. So, the comparative drought stress effect can be measured just as you mentioned. thus, antioxidant activity and proline accumulation are also the predicators of drought stress application, just like photosynthesis or plant chlorophyll fluorescence.

Point 3: Why was instrumental analysis not performed?

Response: Yes, the amount of capsaicin is estimated by organoleptic test because the HPLC facility was not available at that time, but we used a standard method to measure the pungency level used in the previous studies.

Point 4: Change all ml to mL

Response: Replaced ml by mL throughout the MS

Point 5: It must be converted to g force or RCF. Without the full characteristics of the centrifuge rotor, the RPM value adds nothing.

Response: Thanks, it converted to g force (Line 140, and 154)

Point 6: Need to explain how drought stress affected analyzed parameters. Relationship among them may be revealed by, for example, a PCA biplot (Line 203-218).

Response: Thanks for the suggestion; we have added the PCA analysis and bi-plot to explain it

Point 7: The quality of this figure is very low.

Response: Thanks, we have replaced the high-quality figure

Point 8: Section 3.2.3 should be deleted. It is quite obvious that with a water deficit there will be differences.

Response: Thanks for your suggestion but sorry we cannot consider it because other reviewers may not be agreed with it

Point 9: This paragraph should be rewritten. This is not a discussion of the results, but a description of other data. Most of this part of the manuscript should be included in the introduction. Explain where the observed differences come from. Have other authors observed similar changes in studies of other plants?

Response: Thanks, we have revised this paragraph (Line 316-334)

Point 10: Incorrect citation system

Response: Corrected (Line 320)

Round 2

Reviewer 1 Report

I thank the author for their effort to improve the manuscripts. Some of the points were addressed apropriately, however, some points still require some attention.

Point 1: References to recent research on caspaicin synthesis under drough stress was included into introduction. However, this review is under impression that this was done only mechanically - i.e. 82-87 - 'with genetic potential and genotypes' (repetitive), 'total phenols played a significant in pungency' (instead in antioxidant activity) - the sentence from the original manuscript losts its meaning in the shuffling the original wording. 

Point 2: Antioxidant activity - unit of enzyme activity is usually defined as the amount of the enzyme that catalyzes the conversion of one micromole of substrate per minute under the specified conditions of the assay method. The extinction coeficient should be taken into account when converting absorbance changes to enzyme units. Alternatively, one could only give relative increases or decreases in enzyme activity coused by drough which would probably be sufficient and appropriate to support the main conclusions in the manuscript. I would advise against the arbitratry definition of enzyme unit as the change of absorbance of 0.001 Abs/min.

Point 3: Figure 1 is very hard to read (tiny, lacking proper explanation in the caption), some confusion with the numbering of Figures (two Figures 1)

Point 4: Capsaicin content - the supplementary table has been added, however numbers reported in the text unfortunately do not match those shown in the table.

Author Response

Dear Reviewer

Thanks for your valuable comments and help us to improve our MS again

Major comments

There are some major comments that need to be addressed before final assessment of a manuscript. The paper is lacking mention of research done by others who investigated drought stress in pepper and how it relates to capsaicin content and physiological responses in plants. These should be noted in the Introduction and/or discussed in the Discussion section of the manuscript.

Point 1. The paper is lacking mention of research done by others who investigated drought stress in pepper and how it relates to capsaicin content and physiological responses in plants. These should be noted in the Introduction and/or discussed in the Discussion section of the manuscript.

Response: Thanks for valuable suggestions. We have explained that how capsaicin content and physiological responses under drought stress in plants. Additionally, we have discussed the cross talk on effects of drought stress on capsaicin content in pepper, in the introduction section.

Point 2. The paper has several methodological issues. The amount of capsaicin is not determined quantitatively (HPLC or similar) but estimated by organoleptic test, which is a limitation of the study that should be mentioned in the discussion. Proline content was determined by ninhydrin reagent, but it seems without the proline standard was not used to assess the content quantitatively? This should also be noted. How are enzyme units calculated for each antioxidant enzyme and expressed? (Per g of fresh weight?)

Response: Yes, the amount of capsaicin is estimated by organoleptic test because the HPLC facility was not available that time, but we used a standard method to measure the pungency level which has been using in the previous studies.

We re-considered the proline estimation method from 520nm absorbance values by standard curve method.  The figure 4 has revised in MS.  where proline concentration is mentioned as µmoles proline/g of fresh weight.

The unit of enzyme activity has defined as a decrease (CAT and APX) or increase (GPX) in the absorbance of 0.001/min at respective wavelength.

Response: Production traits have been explained in section 2.2

Point 4. Lines 194-197: The explanation in the text seems to contradict the respective Figure 3. (Ghotki better in terms of fruit number than Green wonder or PPE-311?)

Response: The related explanation has been revised to make it clearer and more precise in last three line of section 3.2.2

Response: It has been done

Point 5. Lines 221-222: It is not clear how the sentence relates to the Figure 5, i.e. it contradicts the explanation in 225-227?

Response: Figure 5 has been replaced and explanation also has revised in section 3.2.4

Point 6. Capsaicin contents: There is some data missing??? The text refers to ‘drastic decrease in pungency units’ in DP, but there are no data to document this.

Minor comments

Line 24: 'etc' should not be used here. Please, be specific about the parameters measured.

Response: Addressed

Line 27: '1.1. absorbances at 520 nm' is not a quantitative measure of the proline level and should be omitted from the abstract

Response: Addressed

Line 44: Antioxidant activity was described in the previous sentence, needs not be repeated here.

Response:  In the second sentence we explained the types of phytochemicals so anti-oxidants is one of them

Line 47: It is a kind of overstatement to claim that vitamin C and provitamin A reduce the risk of cancer (and some other diseases); ref. 8? please check if ref. 8 is appropriately cited to support the role in epithelial differentiation or the role in cancer prevention.

Response:  Yes, author appropriately explained it in a review article titled: “vitamin C, and vitamin E as protective antioxidants in human cancers”, published in Annual Review of Nutrition

Line 53: Capsaicin biosynthesis pathway should be described more clearly – there are no two pathways for capsaicin synthesis, instead products of two metabolic pathways are needed for the action of capsaicin synthetase…A reference is missing for the important role of PAL.

Response: Capsaicin biosynthesis pathway has explained more clearly.

Line 104: Please provide full reference for Peterson, 1959. 'For longer fruits, the shape index is greater then one, but is equal to one'. The sentence is not clear.

Response:  Reference added and sentence also has revised

Line 111: Please provide full reference for Reddy and Sasikala, 2013

Response:  Added

Line 189: ‘Water deficiency at the vegetative stage does not affect fruit yield.’ If this is a general knowledge and not the observation form the study, it should be moved from the Results to the Discussion section.

Response: it was misleading statement, it has revised

Figure 4. ‘Sweet pepper’ used in the figure instead of bell pepper

Response: Figure revised

List of references needs to be edited for formatting and consistency.

Response: Addressed

Reviewer 3 Report

Dear Authors,

your revised paper has been sent again for my consideration. I have to admit that the manuscript has been greatly improved. I have a few more minor comments.

  • Centrifugal force should look like this: 8042.4 x g for 10 minutes (x is missing)
  • all µl change to µL
  • L161: (Aebi, 1984) incorrect citation system
  • Please unify units (i.e. L164 'min−1g−1' should be change to '[μg/min x g])
  • Figures 1D and E should be larger as they are unclear

Author Response

Thank you for your time

Point 1. Centrifugal force should look like this: 8042.4 x g for 10 minutes (x is missing)

Response:  Thanks, it has fixed

Point2. All µl change to µL

Response: All µl have changed to µL

Point 3. L161: (Aebi, 1984) incorrect citation system

Response: Thanks, and error has fixed (Line 158)

Point 4. Please unify units (i.e. L164 'min−1g−1' should be change to '[μg/min x g])

Response: Thanks, all have changed (Line 158, 161, and 163)

Point 4. Figures 1D and E should be larger as they are unclear

Response: Fixed